# Evaluating Verifiability in Generative Search Engines

**Nelson F. Liu**[*]     **Tianyi Zhang**     **Percy Liang**
Computer Science Department
Stanford University
nfliu@cs.stanford.edu

## Abstract

Generative search engines directly generate responses to user queries, along with in-line citations. A prerequisite trait of a trustworthy generative search engine is *verifiability*, i.e., systems should cite comprehensively (high citation recall; all statements are fully supported by citations) and accurately (high citation precision; every cite supports its associated statement). We conduct human evaluation to audit four popular generative search engines—Bing Chat, NeevaAI, perplexity.ai, and YouChat—across a diverse set of queries from a variety of sources (e.g., historical Google user queries, dynamically-collected open-ended questions on Reddit, etc.). We find that responses from existing generative search engines are fluent and *appear* informative, but frequently contain unsupported statements and inaccurate citations: on average, a mere 51.5% of generated sentences are fully supported by citations and only 74.5% of citations support their associated sentence. We believe that these results are concerningly low for systems that may serve as a primary tool for information-seeking users, especially given their facade of trustworthiness. We hope that our results further motivate the development of trustworthy generative search engines and help researchers and users better understand the shortcomings of existing commercial systems.

## 1   Introduction

Generative search engines fulfill user information needs by directly generating responses to input queries, along with in-line citations (Figure 1).[1] Existing generative search engines are rapidly gaining users—in March 2023, Microsoft reported that "roughly one third of daily preview users are using

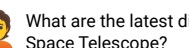

Figure 1: Generative search engines answer user queries by generating a tailored response, along with in-line citations. However, not all generated statements are fully supported by citations (citation recall), and not every citation supports its associated statement (citation precision).

[Bing] Chat daily", and that Bing Chat served 45 million chats in the first month of its public preview (Mehdi, 2023). Generative search engines have the potential to transform how people find information online, but generated responses from existing large language model-backed generative search engines may not always be accurate (Maynez et al., 2020). Given their potential and rapid mainstream adoption, it is critical to evaluate these systems to better understand their potential limitations (akin to prior work in algorithmic auditing; Metaxas and Pruksachatkun, 2017; Buolamwini and Gebru, 2018; Kiritchenko and Mohammad, 2018; Robertson et al., 2018; Metaxa et al., 2019; Green and Chen, 2019; Birhane et al., 2022, *inter alia*).

---

[*]This work would not be possible without the 34 annotators who performed human evaluation; we thank them for their contributions.

[1]In contrast, conventional search engines are limited to retrieving pre-existing webpages.

A prerequisite trait of a trustworthy generative search engine is *verifiability*,[2] that is, each generated statement about the external world should be fully supported by a set of in-line citations, and each provided citation should support its associated statement. Verifiability enables readers to easily check that any generated statement is supported by its cited source.

We conduct a human evaluation to audit four popular commercial generative search engines (Bing Chat, NeevaAI, perplexity.ai, and YouChat) across a diverse set of information-seeking queries (e.g., various types of historical Google user queries from NaturalQuestions (Kwiatkowski et al., 2019), dynamically-collected open-ended questions from Reddit; see Appendix A for examples).

For each query-response pair, we use human evaluation to measure a variety of dimensions:

1. *fluency* (whether the generated text is fluent and cohesive; §2.2);
2. *perceived utility* (whether the generated answer is helpful and informative; §2.2);
3. *citation recall* (the proportion of generated statements about the external world that are fully supported by their citations; §2.3); and
4. *citation precision* (the proportion of generated citations that support their associated statements; §2.4).

A trustworthy generative search engine should achieve high citation recall and precision, indicating that its generated citations are comprehensive (every generated statement is fully supported by citation) and correct (every citation supports its associated statement).

We find that existing generative search engine responses often have high fluency and perceived utility (§4.1), but frequently contain unsupported statements or inaccurate citations (low citation recall and precision; §4.2). On average, merely 51.5% of generated sentences are fully supported with citations (citation recall), and only 74.5% of citations support their associated sentence (citation precision). Furthermore, citation precision is *inversely correlated* with perceived utility ($r = -0.96$); the responses that seem more helpful are often those with inaccurate citations (§4.3). This facade of trustworthiness increases the potential for existing generative search engines to mislead users. For example, in Figure 1, a user with little background

knowledge about the James Webb Space Telescope (motivating a query about its recent discoveries) will likely struggle to identify unsupported statements in the generated response. We hypothesize that citation precision is inversely correlated with perceived utility because generative search engines often copy or closely paraphrase from their cited webpages (§4.4). This improves citation precision because copied text is often supported by the cited webpage, but decreases perceived utility when copied statements are irrelevant to the query or the rest of the generated response.

We make the following contributions: first, we define the citation recall and citation precision evaluation metrics, which aim to encourage the development of systems that cite comprehensively and correctly. Second, we conduct a human evaluation of four popular generative search engines, finding that responses are broadly fluent and appear useful, but frequently contain unsupported statements and inaccurate citations, increasing their potential to mislead users. Third, we observe that perceived utility is inversely correlated with citation precision in existing generative search engines, and hypothesize that this inverse correlation occurs when some systems copy or closely paraphrase from cited webpages. To facilitate further work on developing trustworthy generative search engines, we have released our human evaluation annotations.[3]

## 2   Human Evaluation of Fluency, Perceived Utility, and Verifiability

In this section, we formalize the inputs and outputs of the generative search engines we study, describe the evaluation of fluency and perceived utility, and define and describe the evaluation of citation recall and precision. Citation recall and precision are designed to reward systems that cite comprehensively (i.e., high recall; all statements are fully supported by citations) and accurately (i.e., high precision; every cite supports its associated statement). We also define citation $F_1$, a metric that combines citation precision and citation recall.

### 2.1   Task Formulation

Given a user query $q$ as input, a generative search engine produces a text response $r$, which is a string with embedded in-line citations. For the example in Figure 1, the query $q$ is "What are the latest discov-

---

[2]We adopt the term *verifiability* from the Wikipedia community. Verifiability is a core content policy of Wikipedia: statements must be fully supported by provided sources.

[3]github.com/nelson-liu/evaluating-verifiability-in-generative-search-engines

eries from the James Webb Space Telescope?" and the response $r$ is the string paragraph "The James Webb Space Telescope ... used to study the next interstellar interloper [3].", with embedded citations "[1]", "[2]", and "[3]".

To evaluate citation precision and recall, we first segment the $r$ into a set of $n$ statements $\mathcal{S} = \{s_1, \ldots, s_n\}$. In this work, the segmentation $\mathcal{S}$ is set of sentences in the response $r$. For each statement $s_i \in \mathcal{S}$, we construct a (possibly empty) set $\mathcal{C}_i = \{c_{i,1}, \ldots, c_{i,k}\}$ of $k$ citations associated with the statement $s_i$, where $c_{i,j}$ is the $j$th citation associated with the $i$th response statement. For each citation $c_{i,j}$, we have a URL $u_{i,j}$ and its contents $p_{i,j}$. In this work, $\mathcal{C}_i$ is set of citations that occur in $s_i$ (e.g., for $s_i$ = "Blueberries[1], cherries[2], and grapes[3] grow on trees.[4]", $\mathcal{C}_i = \{[1], [2], [3], [4]\}$).

In practice, a sentence may contain multiple independently-verifiable claims (e.g., conjuncts such as "Cups can be made of glass[1] or plastic[2]."), and a single in-line citation's scope is often ambiguous (e.g., a cite marker after two statements could be interpreted as either supporting both statements, or merely the final one); we leave finer-grained evaluation to future work.

## 2.2 Measuring Fluency and Perceived Utility

To measure response fluency, annotators were shown the user query, the generated response, and the claim "The response is fluent and cohesive". We ask annotators to rate their level of agreement with the claim on a five-point Likert scale from *Strongly Disagree* to *Strongly Agree*. We use a similar process to measure perceived utility, asking annotators to rate their level of agreement with the claim "The response is a helpful and informative answer to the query".

## 2.3 Measuring Citation Recall

Citation recall is the proportion of verification-worthy statements that are fully supported by their associated citations (see Figure 2 for several examples). Thus, computing citation recall requires (i) identifying the verification-worthy statements in a response and (ii) evaluating whether each verification-worthy statement is fully supported by its associated citations.

**Identifying verification-worthy statements.** Given the statements $\mathcal{S}$ in a response $r$, we first ask annotators to remove statements in the response

Figure 2: Examples of calculating citation recall and precision. Citation recall measures the proportion of generated statements that are supported by citations. Citation precision measures the proportion of citations that support their associated statements. Partially-supporting citations only improve citation precision when their associated statement is supported by the union of its citations and no other associated citation fully supports the statement by itself (middle example).

that are not verification-worthy. We take the position that *every* generated statement about the external world is verification-worthy, even those that might seem obvious, trivially true, or "common sense". Generated statements may be incorrect, and statements that seem obvious to some readers may be less than obvious to others (e.g., "The Pope is Catholic"). We believe that systems should aim to provide a source for *all* generated statements about the external world, enabling readers to easily verify *any* statement in a generated response.

In practice, almost all system-generated statements are verification-worthy—notable exceptions include statements about the speaker (the system) itself (e.g., "As a language model, I do not have the ability to ban books.") and questions posed to the user (e.g.,"Would you like to learn more?", generated by systems like Bing Chat and YouChat that are deployed in conversational settings).

**Evaluating whether a verification-worthy statement is fully supported by its associated citations.** Given the verification-worthy statements in a response $r$, annotators evaluate whether each statement is fully supported by its associated cita-

tions (see the sentences of generated response in Figure 1 for examples). To collect these binary judgments, we use the *attributable to identified sources* (AIS) evaluation framework of Rashkin et al. (2022). In particular, a statement $s_i$ is fully supported by its associated citations $\mathcal{C}_i$ if a generic hearer would affirm the statement "According to cited webpages $\mathcal{C}_i$, $s_i$", within the context of the query $q$ and response $r$, and unsupported otherwise.

## 2.4 Measuring Citation Precision

Citation precision is the proportion of generated citations that support their associated statements (Figure 2). In contrast to citation recall, citation precision rewards systems for citing accurately—a response that cites every webpage on the Internet for each generated statement would have high citation recall, but low citation precision (since many articles are irrelevant and do not support their associated statement). To measure citation precision for a response $r$, we first ask annotators to judge whether each citation $c_{i,k}$ contributes full, partial, or no support for its associated statement $s_i$ (see cited webpages in Figure 1 for examples):

- *Full support*: all of the information in the statement is supported by the citation.
- *Partial support*: some of the information in the statement is supported by the citation, but other parts are not supported (e.g., missing or contradictory).
- *No support*: the citation does not support any part of the statement (e.g., the cited webpage is completely irrelevant or contradictory).

For statements that have multiple associated citations, we additionally ask annotators whether the union of its associated cited webpages collectively provides full support for the statement (a binary judgment). Similar to citation recall, we use the AIS evaluation framework of Rashkin et al. (2022) to collect these binary judgments.

To calculate citation precision, let $T_{fs}$ be the number of citations that fully support its associated statement, and let $T_{ps}$ be the number of citations that partially supports its associated statement, where the associated statement is fully supported by the union of its associated citations and no associated citation fully supports the statement by itself.[4] Let $N$ be the total number of citations

in the response. Then, the citation precision is $(T_{fs} + T_{ps})/N$.

## 2.5 Citation $F_1$

Citation $F_1$ is a metric that combines citation precision and citation recall by taking their harmonic mean:

$$F_1 = 2 \cdot \frac{\text{citation precision} \cdot \text{citation recall}}{\text{citation precision} + \text{citation recall}}$$

To achieve a high citation $F_1$, systems must have high citation precision *and* high citation recall.

## 3 Evaluation Setup

In this section, we describe the evaluated generative search engines (§3.1), the diverse query distributions we use for evaluation (§3.2), and the details of our human evaluation protocol (§3.3).

### 3.1 Evaluated Generative Search Engines

We evaluate four existing commercial generative search engines: Bing Chat, NeevaAI, perplexity.ai, and YouChat. [5] These systems pattern after prior work (e.g., Nakano et al., 2021; Menick et al., 2022; Glaese et al., 2022; Thoppilan et al., 2022, *inter alia*) and generate responses by conditioning large language models on the input query and retrieved content (e.g., search results from a conventional search engine). For each input, we save the system's first complete response (i.e., single-turn). Responses were scraped between late February and late March 2023.

Note that evaluated generative search engines have differing abstention rates (Table 1), which can make direct comparison difficult—one might expect that systems with higher abstention rates might also have higher evaluation performance, since they can simply abstain from generating responses to difficult queries (we do not find this to be the case in practice). NeevaAI abstains from responding on nearly 23% of evaluated queries, since its response

---

[4] For an intuitive example of when partially-supporting cites count toward improving precision (greater $T_{ps}$), consider the statement "Health benefits of cycling include improved cardiovascular health[1] and lowered cholesterol levels[2]." with its associated citations [1] and [2]. Suppose that these citations each contribute partial support for the entire statement—the first citation [1] only states that "Health benefits of cycling include improved cardiovascular health", and second citation [2] only states that "Health benefits of cycling include lowered cholesterol levels". Taken together, the citations offer full support for the statement. Although these citations do not fully support the statement on their own, they still meaningfully contribute to its verifiability—systems should not be penalized for aggregating information from multiple citations.

[5] We do not evaluate OpenAI's ChatGPT or Google's Bard because they do not provide in-line citations for their responses (as of March 2023), and thus trivially have low verifiability.

| | Abstention Rate ($\downarrow$) |
|---|---|
| Bing Chat | < 0.5% |
| NeevaAI | 22.7% |
| perplexity.ai | < 0.5% |
| YouChat | < 0.5% |

Table 1: Generative search engines may be designed for deployment in different contexts. NeevaAI abstains from responding to 22.7% of our 1450 queries, since its response is designed for display within a conventional search results page. In contrast, the conversational interface of Bing Chat, and YouChat means that systems must generate a response for nearly every input user query (excepting, e.g., query character length limits).

is displayed within a conventional search engine results page. In contrast, Bing Chat, perplexity.ai, and YouChat respond to almost every user query.

## 3.2 Evaluated Query Distributions

To gain a broader understanding of the strengths and weaknesses of existing commercial generative search engines, we evaluate on a diverse set of queries from a variety of sources (e.g., Google user queries, open-ended Reddit questions, how-to queries) requiring knowledge from several different answer types (e.g., short textual spans, long-form paragraph, lists, or tables). See Appendix A for example queries from each distribution. Each system is evaluated on 1450 queries—150 randomly-sampled queries from each of AllSouls, davinci-debate, ELI5 (KILT / Live), and WikiHowKeywords, and 100 randomly-sampled queries for each of the seven NaturalQuestions subdistributions.

**AllSouls.** We evaluate systems on open-ended essay questions taken from the entrance exam (general paper component) for All Souls College, Oxford University. These questions cover topics including the arts, science, politics, literature, current events, and issues in education and sport.

**davinci-debate.** We evaluate systems on debate topics generated from `text-davinci-003`. To generate debate queries, we follow the procedure of Bakker et al. (2022); see Appendix B.1 for details.

**ELI5.** We take queries from the "Explain Like I'm Five" (ELI5) subreddit, where users provide long-form layperson-accessible answers to submitted questions. Submitted questions are required to admit objective explanations, and answering them often requires long-form textual responses.

We consider two subdistributions of ELI5 queries: ELI5 (KILT) and ELI5 (Live). **ELI5**

**(KILT)** uses historical queries from the KILT ELI5 dataset (Fan et al., 2019; Petroni et al., 2021), drawn from posts created before July 2018. A retrieval-based system could hypothetically perform well on ELI5 (KILT) by simply identifying the query's source Reddit ELI5 post and copying its content. As a result, we also evaluate generative search engines on the **ELI5 (Live)** subdistribution, which increases ecological validity by evaluating systems on real user queries at their time of creation and reducing the incidence of search results with the query's *exact* keywords. [6] We continuously listen to the stream of new Reddit ELI5 posts and immediately query generative search engines for responses whenever a new post is created. This ensures that the source ELI5 post will not have been indexed (and thus, cannot be retrieved) by conventional search engines. minimizing the possibility that the generative search engine has access to the source ELI5 post.

**WikiHowKeywords.** We evaluate systems on queries derived from WikiHow articles. We found that directly querying generative search engines with WikiHow article titles yields responses that largely paraphrase or copy text directly from WikiHow. As a result, we use `text-davinci-003` to paraphrase article titles (e.g., "How to Cut An Avocado") into keyword queries (e.g., "cut avocado").

**NaturalQuestions.** We evaluate generative search engines on NaturalQuestions (Kwiatkowski et al., 2019) queries, stratified by their answer type. NaturalQuestions contains historical queries issued to the Google search engine coupled with long and short answers extracted from Wikipedia. We evaluate on queries from 7 NaturalQuestions subdistributions: queries with paragraph-type long answers (i) with and (ii) without short answers, queries with list-type long answers (iii) with and (iv) without short answer, queries with table-type long answers (v) with and (vi) without short answers, and finally (vii) queries with no long answer (and thus no short answer either).

**Summary.** In total, we evaluate existing generative search engines on 12 total query distributions. Eight query distributions are taken from prior work (ELI5 (KILT) and the seven NaturalQuestions query distributions), while four query distributions were constructed for this work: AllSouls,

---

[6]The goal of the ELI5 (Live) subdistribution is not to identify queries with no relevant webpages on the Internet. We recognize that many user queries express previously-seen information needs, even if the precise wording differs.

davinci-debate, ELI5 (Live), and WikiHowKeywords. These diverse settings provide broad coverage of several potential use cases and information needs, helping us gain a comprehensive understanding of systems' strengths and weaknesses.

### 3.3 Human Evaluation Protocol

**Annotation process.** Evaluating a single query-response pair requires human annotators to complete a three-step The first step measures the response's fluency and perceived utility (§2.2), and the second and third step provide the judgments necessary to measure citation recall (§2.3) and precision (§2.4). See Appendix C for screenshots of the annotation interface and Appendix D for the annotation guidelines.

**Annotator recruitment and training.** Annotation was performed on Amazon Mechanical Turk. Annotators were pre-screened with a qualification study, which required them to read an annotation guidelines document and evaluate five representative query-response pairs. We individually reviewed submitted annotations for qualification study and provided annotators with personalized feedback to help correct any misconceptions or confusion about the task. Annotators who performed well on the qualification study and demonstrated thorough understanding of the task and annotation guidelines were permitted to participate in the main round of human evaluation. We remained in constant contact with annotators throughout the human evaluation process to answer questions about corner-cases and clarify intended behavior. In total, 34 annotators participated in human evaluation.

**Annotator compensation.** Annotators were compensated $1.00 per query-response pair for responses with citations, and $0.38 per query-response pair for responses without citations ($15.00 per hour, by conservative time estimates). On average, annotators took approximately four minutes to complete all three steps for a single query-response pair for responses that contained at least one citation.

**Annotation agreement.** Each query-response pair is annotated once in the human evaluation process. To measure inter-annotator agreement, we collected three annotations for 250 randomly-sampled query-response pairs, finding high agreement rates (greater than 82.0% pairwise agreement and 91.0 F1 for all judgments; see Appendix E).

## 4 Results and Analysis

This section presents the results of our human evaluation study and discusses our main observations and analyses. We see that fluency and perceived utility are generally high across different generative search engines (§4.1), while citation recall and precision are quite low (§4.2), though performance certainly varies by system and query distribution—the low citation recall and precision, when combined with the facade of trustworthiness from fluency and high perceived utility, increase the potential for existing generative search engines to mislead users. Our results also show that citation precision is inversely correlated with perceived utility in existing generative search engines (§4.3). We hypothesize that this is a byproduct of systems' propensity to copy or closely paraphrase text from cited webpages, which may increase citation precision and decrease perceived utility (§4.4).

### 4.1 Fluency and Perceived Utility

See Appendix F for full fluency and perceived utility results for every generative search engine on each of our query distributions.

**Generated responses are fluent and appear helpful.** Averaging across all systems and responses yields an average rating of 4.48 for fluency and 4.50 for perceived utility, indicating that annotators generally found generated responses fluent and helpful for answering the user's input query.

**Comparing fluency and perceived utility between generative search engines.** Comparing fluency and perceived utility ratings between the generative search engines (aggregated over all responses), we see that Bing Chat receives the lowest fluency / perceived utility ratings (4.40 / 4.34), followed by NeevaAI (4.43 / 4.48), perplexity.ai (4.51 / 4.56), and YouChat (4.59 / 4.62).

**Comparing fluency across query distributions.** Comparing average fluency ratings across different query distributions, we see similar ratings between NaturalQuestions queries that have a long answer (i.e., an extractive answer of some length exists on Wikipedia) and non-NaturalQuestions distributions (4.50 vs. 4.47, respectively). Comparing average fluency ratings between NaturalQuestions subdistributions, we see that generated responses to queries that have a short extractive answer are generally more fluent (4.55) than responses to queries with only a long answer (4.46) or those without a long

answer (4.46), perhaps because responses to questions with short answers are generally shorter and often only require factoid knowledge.

A notable outlier distribution is NaturalQuestions queries with table-type long answers and no short answers, where system responses are dramatically less fluent (average of 4.36 across systems vs. average of 4.48 across all query distributions). These challenging queries often require aggregating information across table cells or retrieved sources, since the lack of a short answer implies that no single Wikipedia table cell directly answers the question (e.g., the query "how many grammys does beyonce have without destiny's child"). When the retrieved webpages do not contain a clear extractive answer to the query, but contain facts that seem relevant (e.g., information about Destiny's Child's first Grammy, or Beyonce's total number of career Grammy awards), the generated response is often a stilted agglomeration of statements from various sources, reducing overall fluency.

**Comparing perceived utility across query distributions.** In contrast to fluency, perceived utility can differ substantially between different query distributions. Perceived utility is much higher for NaturalQuestions queries containing a long answer (4.59), as opposed to non-NaturalQuestions queries (4.43). Comparing between different NaturalQuestions subdistributions, we see that perceived utility is highest for queries that have a short answer (4.62), followed by queries that have only a long answer (4.55), and finally by queries that have no long (or short) answer (4.52). Overall, perceived utility decreases as queries require longer-form and less-extractive answers (e.g., factoid NaturalQuestions queries with short answers versus ELI5 queries).

## 4.2 Citation Recall and Precision

See Appendix G for full citation recall and precision results for every generative search engine on each of our query distributions.

**Existing generative search engines often do not cite comprehensively or correctly.** When averaging across all systems, a mere 51.5% of generated statements are fully supported with citations (recall), and only 74.5% of citations fully support their associated statements (precision). We believe these results are unacceptably low for systems that are quickly becoming a popular tool for answering user queries and already have millions of users, especially given that generated responses often *appear* informative and useful.

**Comparing citation recall and precision between generative search engines.** Citation recall and precision varies dramatically between different generative search engines. perplexity.ai achieves the highest average recall (68.7), compared to NeevaAI (67.6), Bing Chat (58.7), and YouChat (11.1). On the other hand, Bing Chat achieves the highest average precision (89.5), followed by perplexity.ai (72.7), NeevaAI (72.0), and YouChat (63.6). A gap of nearly 58% separates the system with the highest and lowest recall (perplexity.ai vs. YouChat), and the gap between the systems with the highest and lowest precision is almost 25% (Bing Chat vs. YouChat).

**Comparing citation recall across query distributions.** Modifying the evaluation query distribution appears to affect citation recall more than citation precision. For example, the gap in citation recall between NaturalQuestions queries with a long answer and non-NaturalQuestions queries is nearly 11% (58.5 vs. 47.8, respectively). Similarly, the difference in citation recall between NaturalQuestions queries with and without short answers is nearly 10% (63.4 for queries with a short answer, 53.6 for queries with only a long answer, and 53.4 for queries with no long or short answer).

We hypothesize that citation recall is driven by the relevance of retrieved webpages. In the absence of retrieved evidence that directly answers the input user query, systems generate statements that are unsubstantiated by citations, resulting in lower recall. For example, generative search engines struggle with citation recall when evaluated on the open-ended AllSouls essay questions (average recall of 44.3), because these queries generally have no extractive answer on the Internet.

**Comparing citation precision across query distributions.** Precision on NaturalQuestions queries with long answers is higher than non-NaturalQuestions distributions (76.1 vs. 72.3, respectively). Precision is highest on NaturalQuestions queries with paragraph answer types (precision of 81.5 when a short answer exists and 78.7 when only a long answer exists). On the other hand, citation precision is lowest when systems are evaluated on AllSouls open-ended essay questions (67.8) and davinci-debate queries (70.3). Comparing between NaturalQuestions subdistributions, average system precision is higher on queries with short answers (77.4) than those with only long answers

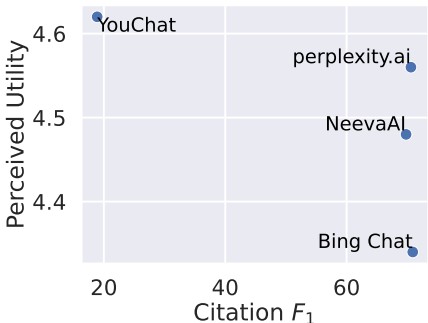

Figure 3: Averaged perceived utility plotted against averaged citation $F_1$ for each evaluated generative search engine. Different systems make different trade-offs between perceived utility and citation $F_1$. Note that these systems are difficult to directly compare since they may have different abstention rates (Table 1).

(74.8) or no long answer (73.5).

**Summary.** To summarize our human evaluation results, Figure 3 plots average perceived utility against average citation $F_1$. Existing systems make different trade-offs between citation recall, citation precision, and perceived utility. See Appendix H for full citation $F_1$ results for every generative search engine on each of our query distributions.

### 4.3 Citation Precision is Inversely Related to Perceived Utility

We find that citation precision is inversely correlated with perceived utility in existing generative search engines ($r = -0.96$). For example, Bing Chat achieves the highest precision, but has the lowest perceived utility. In contrast, YouChat has the lowest citation precision, but its responses attain the highest perceived utility ratings.

This inverse relationship between citation precision and perceived utility is symptomatic of a trade-off between faithfulness and abstractiveness (Ladhak et al., 2022). In particular, we find that system-generated statements often closely paraphrase or directly copy from their associated citations (see §4.4 for further analysis). This results in high citation precision (since extractively copied text is almost always fully supported by the source citation), but lower perceived utility (since the extractive snippets may not actually answer the user's input query). In contrast, systems that frequently deviate from cited content (resulting in low citation precision) may have greater freedom to generate fluent responses that *appear* relevant and helpful to the user's input query.

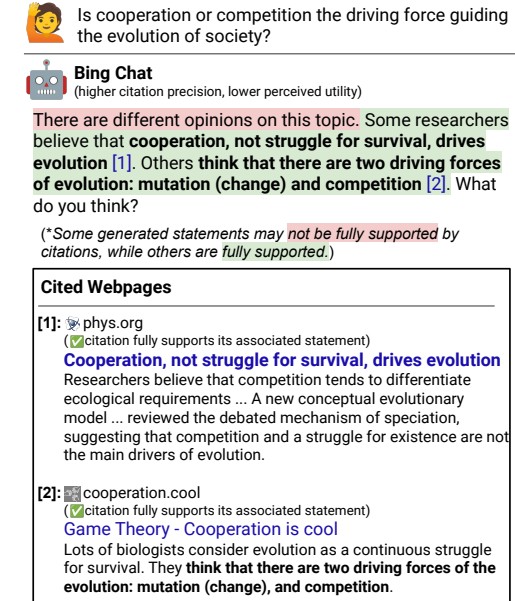

Figure 4: Citation precision is inversely correlated with perceived utility in existing generative search engines. Bing Chat often achieves high citation precision because it closely paraphrases from cited webpages (bolded). However, since these citations are largely irrelevant to the user's input query (biological evolution vs. societal evolution), copying this contents results in lower perceived utility.

This tradeoff is especially apparent on the All-Souls query distribution, which contains open-ended essay questions. AllSouls queries often cannot be answered via extraction from a single webpage on the Internet. For example, given the query "Is cooperation or competition the driving force guiding the evolution of society?", conventional search engine results focus on *biological* evolution, rather than societal evolution. Bing Chat simply copies irrelevant statements directly from the cited sources, resulting in high citation precision but low perceived utility (Figure 4).

### 4.4 Generative Search Engines Closely Paraphrase From Cited Webpages

To better understand how generative search engines use citations to support their responses, we analyze the similarity between generated statements and their supporting cited webpages. For citations that provide full or partial support for their associated statement, annotators were asked to provide *evidence* by copy-pasting the minimal set of sentences from the cited webpage that support their judgment (if any such sentences exist). We compute the BLEU (Papineni et al., 2002) and BERTScore

|              | BLEU | BERTScore (F1) |
|--------------|------|----------------|
| Bing Chat    | 44.1 | 78.8           |
| NeevaAI      | 30.0 | 72.9           |
| perplexity.ai| 22.3 | 69.2           |
| YouChat      | 28.6 | 72.0           |
| Average      | 31.3 | 73.2           |

Table 2: Existing generative search engines closely paraphrase from cited articles; generated statements have high similarity with their cited webpages.

(Zhang et al., 2020) between each generated statement and the annotator-provided evidence from the associated citation. For statements with multiple associated citations, we take the maximum similarity with any associated citation's evidence.

Table 2 presents similarity metrics between generated statements and extracted evidence from supporting webpages—when statements are fully or partially supported by their citations, they often copy or closely paraphrase from their cited articles. Furthermore, systems with higher similarity between their generated statements and cited webpages also have higher average citation precision ($r = 0.80$ between each of BLEU and BERTScore with average citation precision), indicating that their improved precision may largely be a byproduct of their increased tendency to copy or paraphrase from cited webpages.

## 5   Related Work

Existing work has proposed a variety of techniques for building language models that provide references to support generated text. Nakano et al. (2021) use reinforcement learning from human preferences to train language models to answer questions and provide supporting evidence. Similarly, Menick et al. (2022) also use reinforcement learning from human preferences to train language models to answer user questions, but their system generates responses by conditioning on evidence retrieved from a Google search for the given user query. Finally, the LaMDA system of Thoppilan et al. (2022) is trained to provide URLs that support its generated statements. In contrast to the aforementioned line of work on training systems to generate citations, Gao et al. (2022) propose a method for post-editing generated output to reflect and cite retrieved evidence.

Existing work has also proposed evaluation protocols and benchmarks for improving verifiability in language generation systems. Rashkin et al.

(2022) propose the *attributed to identified sources* (AIS) evaluation framework to assess whether a particular statement is supported by provided evidence and validate their guidelines on conversational question answering, summarization, and table-to-text systems. Bohnet et al. (2023) introduce the task of attributed question answering, where systems are given an input question and must output an answer string with a pointer to evidence text supporting the answer, and propose a reproducible evaluation setup with NaturalQuestions queries (only paragraph answer type containing long and short answers) with Wikipedia as the evidence corpus.

In contemporaneous work, Peskoff and Stewart (2023) have domain experts evaluate ChatGPT and YouChat responses to 100 expert-written questions. They find that generated responses are coherent and concise, but frequently undersourced and inaccurate; our results also show that YouChat responses frequently lack citations for generated statements (i.e., low citation recall).

## 6   Conclusion

In this work, we used human evaluation to audit the verifiability of four popular commercial generative search engines—Bing Chat, NeevaAI, perplexity.ai, and YouChat. We find that responses from existing generative search engines are generally fluent and often *appear* informative, but frequently contain unsupported statements and inaccurate citations (low citation recall and precision)—a mere 51.5% of generated statements are fully supported by citations (recall), and only 74.5% of citations support their associated statements (precision). We believe that existing systems' citation recall and precision are unacceptably low, given that they are quickly becoming a popular tool for answering user queries and already have millions of users. Moreover, we find that citation precision is inversely correlated with perceived utility in existing generative search engines—the responses that seem more helpful are often those with more unsupported statements or inaccurate citations. Analysis suggests that this inverse correlation occurs in existing systems because of their propensity to copy or closely paraphrase from cited webpages, which inflates citation precision at the cost of lower perceived utility. We hope our results and insights further motivate the development of trustworthy generative search engines and help researchers and users better understand their current shortcomings.

## Acknowledgements

We are grateful to the 34 annotators who participated in our human evaluation study—this work would not have been possible without them. We also thank Rishi Bommasani, Ge Gao, Natasha Klein-Atlas, Vivian Lai, Kevin Lin, John Thickstun, Eric Wallace, and Gerben Wierda for feedback and discussions that helped improve this work. We thank Amazon Web Services for providing Amazon Mechanical Turk credits that helped support this work. This work was supported in part by the AI2050 program at Schmidt Futures (Grant G-22-63429).

## Limitations

The primary goal of this work was to assess *verifiability* in generative search engine responses. However, note that verifiability is not *factuality*—rather than arbitrating if a generated statement is true (difficult for all but the simplest claims; Rashkin et al., 2022), verifiability enables users to easily check any generated statement's source, allowing them to draw their own conclusions about whether to trust the generated statement. Studying the factuality of generative search engines (that may or may not provide citations) is an important direction for future work—users may not necessarily bother to check the sources, especially given that responses often seem helpful and sound confident, and we'd thus like responses to be as factual as possible.

In our evaluation of verifiability, we consider sentence-level claims. However, sentences often have multiple claims (e.g., "Cats[1] and dogs[2] are common pets."). However, there is currently no clear linguistic definition on what constitutes a claim. As a result, we use sentences for simplicity and reproducibility. Proposing a concrete definition of a "claim" and performing a finer-grained evaluation is an interesting direction for future work.

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

# A  Example queries from each evaluated query distribution

| Source | Example Queries |
|---|---|
| **AllSouls** | What are the functions of fashion? 
 Should wealth be inheritable? |
| **davinci-debate** | Should private companies be allowed to manage public utilities? 
 Should controversial opinions be censored on social media? |
| **ELI5 (KILT)** | Why is a circle 360 degrees and not 100 degrees? 
 Why can animals drink dirty water safely but humans can't? |
| **ELI5 (Live)** | Why jumping into water from great height feels like landing in concrete? 
 where does the deleted data go |
| **WikiHowKeywords** | age paper using tea 
 ways to stop stressing over exam results |
| **NaturalQuestions** 
 **(paragraph long answer, has short answer)** | who wrote the song god your mama and me 
 what is the queen of spain's name |
| **NaturalQuestions** 
 **(paragraph long answer, no short answer)** | where did knock on wood superstition come from 
 what is the use of tap and die |
| **NaturalQuestions** 
 **(list long answer, has short answer)** | what is the most nominated film for the oscars 
 who played guitar on i want you she's so heavy |
| **NaturalQuestions** 
 **(list long answer, no short answer)** | alicia keys if i ain't got you awards 
 is all of florida in the same time zone |
| **NaturalQuestions** 
 **(table long answer, has short answer)** | how many episodes are there in quantum leap 
 what kind of music is red hot chili peppers |
| **NaturalQuestions** 
 **(table long answer, no short answer)** | where does copa airlines fly in the united states 
 michael jordan career high against every nba team |
| **NaturalQuestions** 
 **(no long or short answer)** | what does the x card mean in uno 
 what changes were made when trinidad and tobago gained independence |

Table 3: Example queries from each of the evaluated query distributions. Queries come from diverse sources and require knowledge from a variety of answer types (e.g., short text span, long-form paragraph, list, or table). Each system is evaluated on 1450 queries—150 randomly-sampled queries from each of AllSouls, davinci-debate, ELI5 (KILT), ELI5 (Live), and WikiHowKeywords, and 100 randomly-sampled queries for each of the seven NaturalQuestions subdistributions.

# B  Query Distribution Details

## B.1  davinci-debate

We seed the data generation process with 100 debate questions, which are manually transformed propositions propositions taken from the Perspectrum dataset of Chen et al. (2019) (e.g., the proposition "Vaccination must be made compulsory." could be rewritten as the question "Should vaccines be mandatory?").

To generate a debate question, we prompt text-davinci-003 with 10 randomly-sampled seed questions. We repeat this procedure until we have generated 150 unique debate questions that also do not appear in our seed set. Finally, generated questions were manually filtered for inappropriate content.

## B.2  ELI5

To transform ELI5 post titles into queries, we remove ELI5-specific prefixes (e.g., the post title "ELI5: why can't our brains recall every memory?" becomes the query "Why can't our brains recall every memory?").

## B.3  WikiHowKeywords

To paraphrase article titles into keyword queries, we prompt text-davinci-003 with "Given a question, write a concise Google search query that would answer the question" and two in-context examples.

## C  Annotation Interface

Figures 5-7 show the annotation interface used for human evaluation.

In the first step, annotators were shown the query and the generated response (without citations) and asked to rate response fluency and perceived utility on a five-point Likert scale.

In the second step, annotators were shown the statements in the generated response (including any generated citations) and asked to filter out statements are not verification-worthy.

Finally, in the third step, annotators were shown the statements that were previously judged to require verification (in the prior step), as well as each statement's associated system-generated citations. For each statement and associated citation, annotators judged whether the citation fully supports, partially supports, or does not support the statement, as interpreted within the broader context of the query and system response. For statements with multiple associated citations, annotators are asked to judge whether the citations, when taken together, fully support the statement; this captures cases where multiple citations support disjoint parts of a statement (e.g., "Health benefits of cycling include improved cardiovascular health[1] and lowered cholesterol levels[2].").

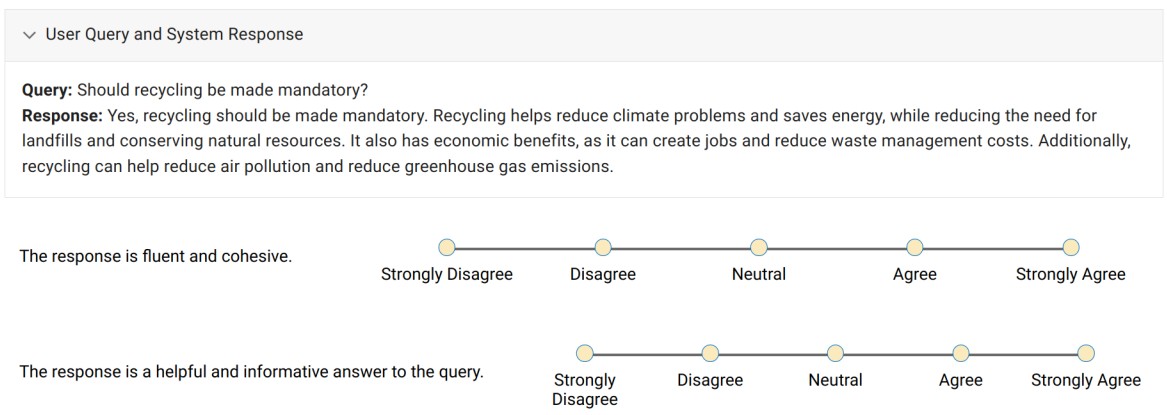

Figure 5: First step of the annotation interface, where annotators judge response fluency and perceived utility.

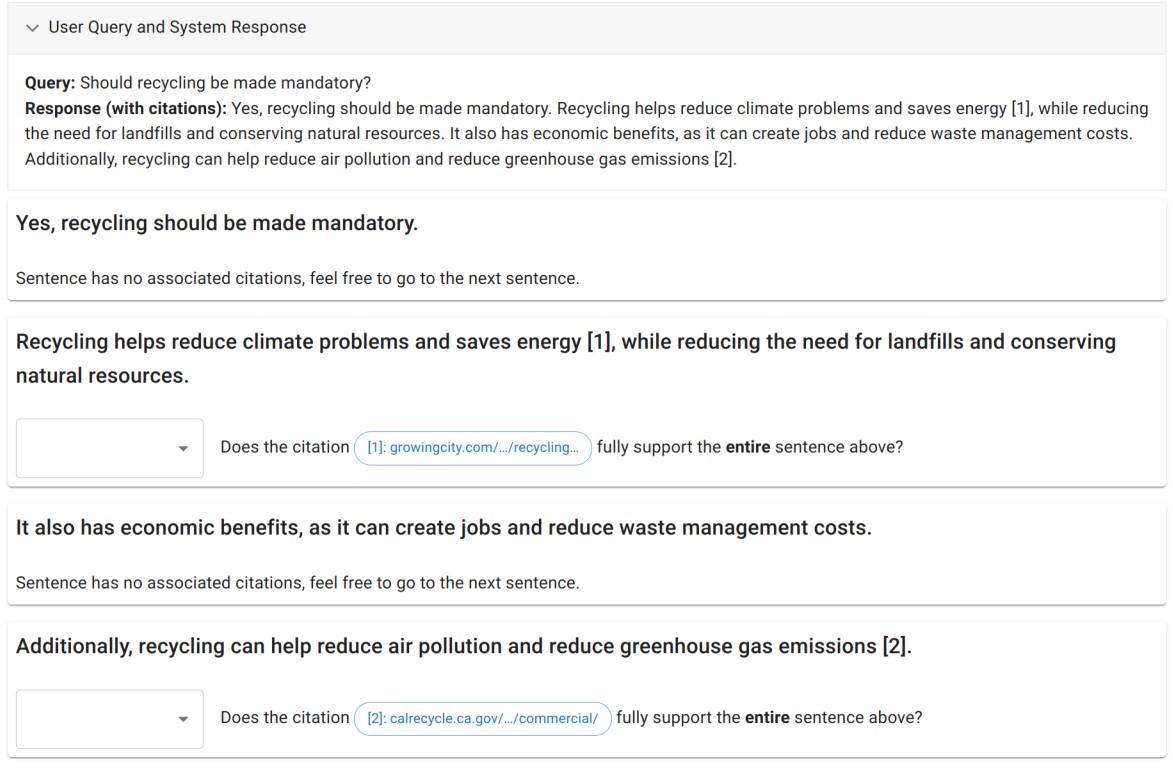

> ＞ User Query and System Response

☑ Yes, recycling should be made mandatory.

☐ Recycling helps reduce climate problems and saves energy [1], while reducing the need for landfills and conserving natural resources.

☑ It also has economic benefits, as it can create jobs and reduce waste management costs.

☐ Additionally, recycling can help reduce air pollution and reduce greenhouse gas emissions [2].

Figure 6: Second step of the annotation interface, where annotators uncheck statements that are not verification-worthy. Statements that contain generated citations must be verification-worthy, so we automatically mark them as such in the interface (greyed-out checkboxes next to the 2nd and 4th sentences above).

> ﹀ User Query and System Response

**Query:** Should recycling be made mandatory?
**Response (with citations):** Yes, recycling should be made mandatory. Recycling helps reduce climate problems and saves energy [1], while reducing the need for landfills and conserving natural resources. It also has economic benefits, as it can create jobs and reduce waste management costs. Additionally, recycling can help reduce air pollution and reduce greenhouse gas emissions [2].

**Yes, recycling should be made mandatory.**

Sentence has no associated citations, feel free to go to the next sentence.

**Recycling helps reduce climate problems and saves energy [1], while reducing the need for landfills and conserving natural resources.**

[ ▾ ]   Does the citation ( [1]: growingcity.com/.../recycling... ) fully support the **entire** sentence above?

**It also has economic benefits, as it can create jobs and reduce waste management costs.**

Sentence has no associated citations, feel free to go to the next sentence.

**Additionally, recycling can help reduce air pollution and reduce greenhouse gas emissions [2].**

[ ▾ ]   Does the citation ( [2]: calrecycle.ca.gov/.../commercial/ ) fully support the **entire** sentence above?

Figure 7: Third step of the annotation interface, where annotators provide judgments on whether each citation supports its associated statement, and whether each statement is supported by the union of its citations (only applicable when a statement has multiple associated citations).

## D    Annotation Guidelines

Figures 8-12 show the annotation guidelines we used for the task. We ask crowd annotators to read these guidelines as part of the qualification study. Only annotators that demonstrated a thorough understanding of the guidelines and task were permitted to participate in the main round of human evaluation.

In this task, you will evaluate an AI system's response to a user query. The AI system outputs a paragraph that contains information relevant to the user's query, and we would like to evaluate whether the AI system can accurately cite sources for statements it makes about the external world.

At a high level, this task breaks down into three steps:

1. Evaluating response quality
2. Filtering sentences that do not require citation.
3. Judging whether each statement is fully supported by its citation(s).

**Please carefully read the guidelines below before starting on the task. The task compensation accounts for the time needed to read the guidelines.**

## Preliminaries: Logging In

When first entering the site, you will be prompted to select a username. **Please use your worker ID as the username, so we can keep track of the examples you've annotated.** The top of the interface displays your worker ID, the total number of examples submitted from this username, and will show a completion code when you have finished the task.

If something is wrong with the example, you may press the "Flag Example" button in the top-right corner to report the error. Please do not submit annotations for such examples.

**Your task ends after you've completed 5 responses. A completion code will appear at the top of the interface---there is no need to complete more than 5 responses to receive credit for the study.**

## Step 1: Evaluating response quality

You will be shown the user's original query, and the system's response to the query---please carefully read both of them. Then, you will be asked to rate your level of agreement with two questions:

1. The response is fluent and cohesive.
2. The response is a helpful and informative answer to the query.

Figure 8: First page of the annotation guidelines.

Once you have finished selecting a response for each of the two questions, press the "Next Step" button in the top-right corner to continue.

## Step 2: Filtering sentences that do not require citation.

The goal of this step is to filter the sentences in the system response by removing sentences that do not require citation (unchecking them in the interface). **We expect the majority of sentences produced by the system to require citation, so don't worry if you find yourself rarely unchecking sentences.**

In general, we take the position that **all statements about the external world require citation**, even if they are trivially true or "common sense" (since users may differ in their background, which affects their basic beliefs). For example, the following sentences require citation:

- (1a): The House of Lords is a topic of ongoing debate in the UK.
- (1b): However, there is still no consensus on what should replace the Electoral College.
- (1c): The sky is blue.
- (1d): The moon landing was staged.
- (1e): In February 2023, LeBron James took 261,960 total breaths.
- (1f): Patrick Henry once said "Give me liberty, or give me death".
- (1g): Thanksgiving dinners usually taste bad.
- (1h): Voting rights are controversial

In particular, note that sentences can require citation despite being nearly impossible to verify. Consider example (e) above. It's highly unlikely that anyone knows exactly how many breaths LeBron James took in February 2023, let alone that such information could be linked to in a citation. However, it's still a statement about the external world, and it's still *possible* to find out for certain whether the statement is true or false. Thus, the statement requires citation.

In contrast, consider the following examples of sentences that do not require citation:
- (2a): I believe that the moon landing was staged.
    - **Explanation**: In general, all sentences pertaining to "I" do not require citation. This statement expresses a belief held by the speaker. The speaker is unknown, so this statement does not require citation. Note that the similar-looking statement "The moon landing was staged" (example 1d) require citation and is verifiable.
- (2b): Have you listened to that song?
    - **Explanation**: Questions do not have information to verify.
- (2c): Pick up the ball on the floor.
    - **Explanation**: Commands do not have information to verify.
- (2d): It is the year 2300. Robots rule the earth.

Figure 9: Second page of the annotation guidelines.

- **Explanation**: the sentence "Robots rule the earth." does not require citation, since the context ("It is the year 2300") specifies that this is a hypothetical situation and not a statement about the external world.

Carefully read each sentence again and decide whether it requires citation. If it does not require citation, uncheck its corresponding checkbox. When you have finished, press the "Next Step" button in the top-right corner to proceed.

# Step 3: Judging whether each statement is fully supported by its citation(s).

In this step, you will evaluate whether each statement is supported by its corresponding citations. Note that the system responses may appear very fluent and well-formed, but contain slight inaccuracies that are not easy to discern at first glance. Pay close attention to the text. Read it carefully as you would when proofreading.

Carefully read the user query and the statement. You may also have to re-read the full system response to understand the statement in its full context. Given the statement's associated citations, your task is to judge whether *all of the information provided by the system response is fully supported by the source document*.

In particular, this question can be answered by considering:
- (A): What is the information provided by the statement?
- (B): According to the citation(s), is this statement true?

**(A): What is the information provided by the statement?**

To determine the information provided by the statement, you must consider the query, the statement, and the context of the statement within the full response. The citations should be completely ignored when determining "the information provided by the statement."

Consider the following example:

**Query:** Why do so many people want to get married?
**Response (statement highlighted):** People get married for many reasons, including love, companionship, financial security, and to share their lives with a partner. Marriage can also be seen as a way to affirm mutual love or start a family.

In this case, the statement is stand-alone, and can be interpreted without looking at the query or the rest of the response. However, this is not always the case. Consider the example below:

**Query**: Is it wrong to exaggerate in a letter of recommendation?

Figure 10: Third page of the annotation guidelines.

**Response (statement highlighted):** Yes, it is wrong. Letters of recommendation should reflect the author's honest perspective on the candidate.

The response "Yes, it is wrong" is uninterpretable on its own, because it is not clear what "it" refers to. However, by using the context of the query, it becomes clear that the statement is equivalent to "Yes, [exaggerating in a letter of recommendation] is wrong".

For another example, consider:

**Query:** how many characters are in the prologue of canterbury tales
**Response (statement highlighted):** In Geoffrey Chaucer's The Canterbury Tales, 32 characters make the journey to Canterbury. This includes the narrator, the host, and the Canon's yeoman, who join the group later.

The statement "This includes the narrator, the host, and the Canon's yeoman, who join the group later." is uninterpretable on its own, because it is not clear what "This" refers to, or what "group" they join. The preceding sentence of the response is essential for realizing that this sentence is equivalent to "[The 32 characters that make the journey to Canterbury] include the narrator, the host, and the Canon's yeoman, who join the [32 characters] later".

In general, use your best judgment to determine the information provided by the system response.

### (B): According to the citation(s), is this statement true?

Again, you should use your best judgment in determining whether all of the information provided by the statement is supported by the associated citation(s).

It may be helpful to ask yourself whether it is accurate to say "according to the citation" with a statement following this phrase. For example, is it accurate to say "according to the citation, in Geoffrey Chaucer's The Canterbury Tales, 32 characters make the journey to Canterbury"?

Be sure to check **all** of the information in the statement. You will be given six options:
-   "**Full Support**": All of the information in the statement is supported in the document.
-   "**Partial Support**": Only some of the information is supported in the document, but other parts of the information are missing from the document.
-   "**No Support**": This document does not support any part of the statement.
-   "**Article Not Accessible**": Not able to access the document (e.g., paywall or the link is dead)
-   "**Citation Has Support but also Refutes Statement**": The citation has information that supports the statement, but also has information that refutes the statement.
-   "**Statement is Unclear, Can't Make Judgment**": The statement is so incomprehensible that it cannot be determined if the citation supports the statement.

Figure 11: Fourth page of the annotation guidelines.

**If the citation offers "full support" or "partial support"** of a document, you will also be asked to copy and paste the minimal set of sentences from the article that support your judgment. In cases where you can't localize the judgment to particular sentence(s) (e.g., the entire article supports the statement, or the support comes from an image or graphic), feel free to leave this input blank.

**When a statement has more than one associated citation**, you will also judge whether the citations, when taken together, fully support the statement (**Yes/No**). In other words, if you merged all of these citations into one big webpage (and it became a single citation), would this citation fully support the statement? If the citations contradict each other (e.g., one fully supports the statement, whereas another refutes the statement), then select "**Citations Contradict Each Other**".

Figure 12: Fifth page of the annotation guidelines.

# E  Annotation Quality

Table 4 presents inter-annotator agreement statistics, computed on a random sample of 250 query-response pairs that received annotations each. We measure the pairwise agreement between individual pairs of ratings and an F1 score comparing individual ratings to the majority consensus. We compute agreement on judgments of (i) fluency and perceived utility, (ii) whether a statement is verification-worthy, (iii) whether a citation supports its associated statement, and (iv) whether a statement is fully supported by the union of its citations (in the case where multiple webpages are cited). When calculating agreement on fluency and perceived utility judgments, we coarsen the 5-point Likert judgments into three options: "Disagree", "Neutral", and "Agree". Agreement rates between annotators are high (pairwise agreement greater than 82.0% and F1 greater than 91.0 for all judgments).

| *Inter-Annotator Agreement* (↑) | | |
| --- | --- | --- |
| | Pairwise Agreement % | F1 |
| Fluency | 88.5 | 94.1 |
| Perceived Utility | 86.4 | 93.1 |
| Verifiability | 94.6 | 97.3 |
| Citation Supports | 82.0 | 91.0 |
| Statement Supported | 82.2 | 91.1 |

Table 4: Inter-annotator agreement statistics. Pairwise Agreement % computes the proportion of individual judgment pairs that agree, and F1 compares individual judgments to the majority consensus judgment. Inter-annotator agreement is high (greater than 82.0% pairwise agreement % and 91.0 F1 for all judgments).

# F   Fluency and Perceived Utility

Table 5 presents the fluency of generative search engine responses on each of our query distributions, and Table 6 presents the perceived utility.

*Fluency (↑)*

|  | Average Over All Queries |
|---|---|
| Bing Chat | 4.40 |
| NeevaAI | 4.43 |
| perplexity.ai | 4.51 |
| YouChat | 4.59 |
| Average | 4.48 |

*Fluency (↑)*

|  | AllSouls | davinci-debate | ELI5 KILT | ELI5 Live | WikiHowKeywords |
|---|---|---|---|---|---|
| Bing Chat | 4.31 | 4.37 | 4.36 | 4.30 | 4.41 |
| NeevaAI | 4.50 | 4.53 | 4.50 | 4.42 | 4.42 |
| perplexity.ai | 4.43 | 4.54 | 4.55 | 4.47 | 4.45 |
| YouChat | 4.58 | 4.65 | 4.56 | 4.53 | 4.52 |
| Average | 4.45 | 4.52 | 4.49 | 4.43 | 4.45 |

*Fluency (↑)*

|  | NaturalQuestions List Long Answer Has Short | List Long Answer No Short | Table Long Answer Has Short | Table Long Answer No Short | Paragraph Long Answer Has Short | Paragraph Long Answer No Short | No Answer |
|---|---|---|---|---|---|---|---|
| Bing Chat | 4.49 | 4.52 | 4.46 | 4.30 | 4.54 | 4.41 | 4.39 |
| NeevaAI | 4.45 | 4.40 | 4.31 | 4.28 | 4.41 | 4.49 | 4.43 |
| perplexity.ai | 4.69 | 4.54 | 4.59 | 4.41 | 4.73 | 4.43 | 4.37 |
| YouChat | 4.65 | 4.56 | 4.60 | 4.45 | 4.66 | 4.69 | 4.64 |
| Average | 4.57 | 4.50 | 4.49 | 4.36 | 4.58 | 4.50 | 4.46 |

Table 5: Human evaluation results for generated response fluency (five-point Likert ratings). In general, existing generative search engines produce fluent text. Performance is notably lower on NaturalQuestions queries with table-type long answers and no short answers, which often require aggregating information within or across citations.

*Perceived Utility (↑)*

| | Average Over All Queries |
|---|---|
| Bing Chat | 4.34 |
| NeevaAI | 4.48 |
| perplexity.ai | 4.56 |
| YouChat | 4.62 |
| Average | 4.50 |

*Perceived Utility (↑)*

| | AllSouls | davinci-debate | ELI5 KILT | ELI5 Live | WikiHowKeywords |
|---|---|---|---|---|---|
| Bing Chat | 4.15 | 4.19 | 4.19 | 4.09 | 4.37 |
| NeevaAI | 4.44 | 4.39 | 4.54 | 4.46 | 4.42 |
| perplexity.ai | 4.39 | 4.60 | 4.54 | 4.50 | 4.51 |
| YouChat | 4.53 | 4.54 | 4.53 | 4.50 | 4.63 |
| Average | 4.38 | 4.43 | 4.45 | 4.39 | 4.48 |

*Perceived Utility (↑)*

| | NaturalQuestions List Long Answer Has Short | No Short | Table Long Answer Has Short | No Short | Paragraph Long Answer Has Short | No Short | No Answer |
|---|---|---|---|---|---|---|---|
| Bing Chat | 4.63 | 4.49 | 4.49 | 4.47 | 4.53 | 4.40 | 4.38 |
| NeevaAI | 4.65 | 4.57 | 4.43 | 4.38 | 4.43 | 4.63 | 4.49 |
| perplexity.ai | 4.71 | 4.61 | 4.60 | 4.55 | 4.77 | 4.58 | 4.50 |
| YouChat | 4.72 | 4.64 | 4.70 | 4.54 | 4.77 | 4.77 | 4.70 |
| Average | 4.68 | 4.58 | 4.55 | 4.49 | 4.62 | 4.60 | 4.52 |

Table 6: Human evaluation results for perceived utility of generated responses (five-point Likert ratings). In general, responses from existing generative search engines *appear* informative and useful.

# G  Citation Recall and Precision

Table 7 presents generative search engine citation recall across the evaluated query distributions, and Table 8 presents citation precision.

*Citation Recall (%; ↑)*

|  | Average Over All Queries |
|---|---|
| Bing Chat | 58.7 |
| NeevaAI | 67.6 |
| perplexity.ai | 68.7 |
| YouChat | 11.1 |
| Average | 51.5 |

*Citation Recall (%; ↑)*

|  | AllSouls | davinci-debate | ELI5 | | WikiHowKeywords |
|---|---|---|---|---|---|
|  |  |  | KILT | Live |  |
| Bing Chat | 55.6 | 57.1 | 59.8 | 59.9 | 50.7 |
| NeevaAI | 55.3 | 66.3 | 66.6 | 61.6 | 72.5 |
| perplexity.ai | 63.0 | 64.2 | 64.8 | 58.1 | 74.6 |
| YouChat | 3.2 | 3.9 | 3.0 | 4.6 | 12.1 |
| Average | 44.3 | 47.9 | 48.5 | 46.0 | 52.5 |

*Citation Recall (%; ↑)*

|  | NaturalQuestions | | | | | | |
|---|---|---|---|---|---|---|---|
|  | List Long Answer | | Table Long Answer | | Paragraph Long Answer | | No Answer |
|  | Has Short | No Short | Has Short | No Short | Has Short | No Short |  |
| Bing Chat | 74.1 | 60.6 | 63.5 | 49.2 | 72.1 | 66.3 | 61.9 |
| NeevaAI | 73.0 | 64.2 | 69.5 | 65.1 | 75.0 | 74.8 | 65.6 |
| perplexity.ai | 85.3 | 74.4 | 79.6 | 62.4 | 84.9 | 75.9 | 68.4 |
| YouChat | 21.6 | 16.6 | 30.6 | 11.5 | 31.6 | 21.8 | 17.8 |
| Average | 63.5 | 53.9 | 60.8 | 47.1 | 65.9 | 59.7 | 53.4 |

Table 7: Human evaluation results of citation recall in existing generative search engines. Citation recall is concerningly low (many generated statements are not fully supported by citations), especially given that these systems already have millions of users and may serve as a primary tool for fulfilling user information needs.

*Citation Precision (%; ↑)*

|  | Average Over All Queries |
|---|---|
| Bing Chat | 89.5 |
| NeevaAI | 72.0 |
| perplexity.ai | 72.7 |
| YouChat | 63.6 |
| Average | 74.5 |

*Citation Precision (%; ↑)*

|  | AllSouls | davinci-debate | ELI5 | | WikiHowKeywords |
|---|---|---|---|---|---|
|  |  |  | KILT | Live |  |
| Bing Chat | 88.8 | 88.8 | 87.6 | 87.2 | 92.1 |
| NeevaAI | 69.8 | 74.1 | 75.7 | 73.8 | 74.0 |
| perplexity.ai | 61.7 | 68.4 | 64.9 | 66.3 | 77.4 |
| YouChat | 51.1 | 50.0 | 64.7 | 57.9 | 71.1 |
| Average | 67.8 | 70.3 | 73.2 | 71.3 | 78.7 |

*Citation Precision (%; ↑)*

|  | NaturalQuestions | | | | | | |
|---|---|---|---|---|---|---|---|
|  | List Long Answer | | Table Long Answer | | Paragraph Long Answer | | No Answer |
|  | Has Short | No Short | Has Short | No Short | Has Short | No Short |  |
| Bing Chat | 86.8 | 86.8 | 89.0 | 92.5 | 92.9 | 91.3 | 90.8 |
| NeevaAI | 73.2 | 67.6 | 67.1 | 64.2 | 73.4 | 76.5 | 70.8 |
| perplexity.ai | 82.1 | 81.0 | 76.0 | 71.7 | 83.8 | 79.7 | 74.0 |
| YouChat | 63.3 | 62.7 | 64.8 | 56.1 | 75.7 | 67.5 | 58.6 |
| Average | 76.4 | 74.5 | 74.2 | 71.1 | 81.5 | 78.7 | 73.5 |

Table 8: Human evaluation results of citation precision in existing generative search engines. Citation precision is concerningly low (many generated citations do not support their associated statements), especially given that these systems already have millions of users and may serve as a primary tool for fulfilling user information needs.

# H Citation $F_1$

Table 9 presents the citation $F_1$ for every evaluated generative search engine on each query distribution.

*Citation $F_1$ ($\uparrow$)*

| | Average Over All Queries |
|---|---|
| Bing Chat | 70.9 |
| NeevaAI | 69.8 |
| perplexity.ai | 70.6 |
| YouChat | 18.9 |
| Average | 57.6 |

*Citation $F_1$ ($\uparrow$)*

| | AllSouls | davinci-debate | ELI5 | | WikiHowKeywords |
|---|---|---|---|---|---|
| | | | KILT | Live | |
| Bing Chat | 68.4 | 69.5 | 71.1 | 71.0 | 65.4 |
| NeevaAI | 61.7 | 70.0 | 70.8 | 67.1 | 73.2 |
| perplexity.ai | 62.3 | 66.2 | 64.8 | 62.0 | 76.0 |
| YouChat | 6.0 | 7.2 | 5.6 | 8.5 | 20.7 |
| Average | 49.6 | 53.2 | 53.1 | 52.2 | 58.8 |

*Citation $F_1$ ($\uparrow$)*

| | NaturalQuestions | | | | | | |
|---|---|---|---|---|---|---|---|
| | List Long Answer | | Table Long Answer | | Paragraph Long Answer | | No Answer |
| | Has Short | No Short | Has Short | No Short | Has Short | No Short | |
| Bing Chat | 79.9 | 71.4 | 74.1 | 64.2 | 81.2 | 76.8 | 73.6 |
| NeevaAI | 73.1 | 65.9 | 68.3 | 64.6 | 74.2 | 75.7 | 68.1 |
| perplexity.ai | 83.7 | 77.5 | 77.8 | 66.7 | 84.3 | 77.7 | 71.1 |
| YouChat | 32.2 | 26.2 | 41.5 | 19.2 | 44.6 | 32.9 | 27.4 |
| Average | 67.2 | 60.2 | 65.4 | 53.7 | 71.1 | 65.8 | 60.0 |

Table 9: Citation $F_1$ of generated responses.