# OpenReview forum: "Evaluating Verifiability in Generative Search Engines"
_EMNLP/2023/Conference — EMNLP 2023 Findings_

### Official Review · Reviewer_hMXQ · 2023-08-01

**Soundness:** 4

**Excitement:**

4: Strong: This paper deepens the understanding of some phenomenon or lowers the barriers to an existing research direction.

**Paper Topic And Main Contributions:**

This paper makes an important step towards attribution evaluation in generative search engines by defining citation metrics and comprehensive human evaluations.

Comprehensive human evaluation on the citation of generative search engines.

Important insights and conclusions are found in this paper.

**Reasons To Accept:**

1. Systematic metric definition and comprehensive evaluation metrics and generative search engines.
2. These findings are insightful and shed some lights into developing future generative search engines.
3. This paper is very well-written and easy-to-follow.

**Reasons To Reject:**

1. These conclusion may be outdated quickly due to the fact that these generative search engines being tested may updated quickly.

**Reproducibility:**

5: Could easily reproduce the results.

**Reviewer Confidence:**

4: Quite sure. I tried to check the important points carefully. It's unlikely, though conceivable, that I missed something that should affect my ratings.

**Typos Grammar Style And Presentation Improvements:**

[omitted.link](http://omitted.link) doesn’t work in the footnote of Page 2.

Due to the limited access, I understand we know very little about the model details. That being said, I personally would love to see more explanations (or educated guesses) about these models and their performances.

---

> ### Author Rebuttal · Authors · 2023-08-23
>
> Thanks for taking the time to review our paper. See below for some inline responses:
>
> > These conclusion may be outdated quickly due to the fact that these generative search engines being tested may updated quickly.
>
> Our goal was not to make conclusions about the absolute performance of these systems, but just to show that these systems (in their current instantiation) do not cite reliably and to provide an evaluation blueprint for measuring and tracking progress in this space.
>
> > Due to the limited access, I understand we know very little about the model details. That being said, I personally would love to see more explanations (or educated guesses) about these models and their performances.
>
> Similar to the point above, we wanted to avoid speculating about what's going on underneath the hood of these systems---the point of the evaluation numbers is less to make concrete recommendations about model development (since we don't know how these models work), but to make a sociotechnical contribution and underscore that these (widely used) systems, in their current form, do not provide reliable citations with their responses. As more open-source alternatives for these generative search engines emerge, we hope that the evaluation metrics and process laid out in this paper will provide a useful blueprint for measuring system quality.

---

### Official Review · Reviewer_XbQo · 2023-08-04

**Soundness:** 4

**Excitement:**

3: Ambivalent: It has merits (e.g., it reports state-of-the-art results, the idea is nice), but there are key weaknesses (e.g., it describes incremental work), and it can significantly benefit from another round of revision. However, I won't object to accepting it if my co-reviewers champion it.

**Paper Topic And Main Contributions:**

This paper delves into the issue of verifiability concerning answers produced by Generative Search Engines (GSEs). The primary focus is on analyzing the current search engines' performance using citation recall and citation precision as evaluation metrics, which are notable contributions of the paper. Additionally, the authors undertake a human evaluation of four search engines, revealing instances where the answers provided include unsupported or inaccurate citations. Furthermore, the paper explores the relationship between fluency and perceived utility, which are two additional performance metrics, and their inverse correlation with the citation recall and precision metric. This observation sheds light on the trade-offs between these aspects of GSE-generated answers. Finally, the authors make their human evaluation annotations accessible.


**Reasons To Accept:**

1. Metric Introduction: The authors introduce novel evaluation metrics specifically designed for assessing the quality of answers generated by Generative Search Engines (GSEs). These metrics are essential for accurately measuring and understanding the performance of GSEs in providing verifiable answers.
2. Human Annotations Resource: The paper provides a valuable resource by offering human annotations that assess the quality of answers based on four different metrics. This dataset enhances the research community's understanding of GSEs' capabilities and limitations.
3. Comprehensive Analysis: The authors thoroughly analyze the results of their study, revealing that the answers produced by GSEs exhibit low performance on citation recall and precision metrics. Furthermore, their investigation demonstrates a negative correlation between these metrics and the aspects of fluency and perceived utility in the answers. This analysis yields important insights into the trade-offs between different performance aspects of GSEs' answer generation.

Overall, the introduction of novel evaluation metrics, the provision of a human annotations resource, and the comprehensive analysis of the study's findings make this paper a valuable addition to the field of natural language processing and worthy of acceptance.

**Reasons To Reject:**

The reason to reject this paper is that the authors fail to provide a clear and convincing explanation for why they consider the results obtained on citation precision and recall metrics as unacceptable for Generative Search Engines. Without a solid justification, the claims made in the paper lack credibility and undermine the significance of the research.


**Reproducibility:**

4: Could mostly reproduce the results, but there may be some variation because of sample variance or minor variations in their interpretation of the protocol or method.

**Reviewer Confidence:**

4: Quite sure. I tried to check the important points carefully. It's unlikely, though conceivable, that I missed something that should affect my ratings.

---

> ### Author Rebuttal · Authors · 2023-08-23
>
> Thanks for your review and thoughtful comments. See below for some inline responses:
>
> > The reason to reject this paper is that the authors fail to provide a clear and convincing explanation for why they consider the results obtained on citation precision and recall metrics as unacceptable for Generative Search Engines. Without a solid justification, the claims made in the paper lack credibility and undermine the significance of the research.
>
> We will add additional discussion to the paper. To recap, our human evaluation indicates that, "when averaging across all systems, a mere 51.5% of generated statements are fully supported with citations (recall), and only 74.5% of citations fully support their associated statements (precision)" (lines 465-469 of the paper).
>
> We argue that these citation precision and recall numbers are unacceptable for systems that are quickly being deployed to users and have the responsibility of keeping them informed---given that we know that these systems hallucinate and output incorrect statements, it's difficult to trust these systems knowing that only 51% of statements can be cross-referenced with accurate citations, and that 25% of citations flat-out do not support their associated claims.
>
> In fact, Bing.com recently started to display generated output to users of their conventional search engine by default, massively broadening its usage and exposure. The overwhelming majority of these users have no expertise with language models and thus have no understanding of where they might fail---they see informative-looking text on the internet (with citations, generally a hallmark of well-researched response) and automatically assume that the citations support their corresponding statements.
>
> Our work quantifies generative search engine trustworthiness (via citation precision and recall), which enables researchers to track and improve these systems.

---

### Official Review · Reviewer_qPBo · 2023-08-10

**Soundness:** 2

**Excitement:**

2: Mediocre: This paper makes marginal contributions (vs non-contemporaneous work), so I would rather not see it in the conference.

**Paper Topic And Main Contributions:**

This paper reports the results of a human audit of the verifiability of responses by popular generative search engines - Bing Chat, NeevaAI, perplexity.ai and YouChat. The authors define new recall and precision metrics for the citations produced by generative search engines. The key finding is that, despite generally high fluency and perceived utility of generative search engine responses, only 51.5% of them are fully supported with citations, whereas 74.5% of citations actually support their associated responses. A secondary finding is that citation recall and precision are inversely correlated with fluency and the perceived utility of responses.

**Reasons To Accept:**

* No other existing studies of generative search engines
* Findings that may help improve the quality of responses by generative search engines

**Reasons To Reject:**

* Weaknesses in the definition of citation precision: although all citations for a single response are evaluated by a single worker, the citations are evaluated individually rather than holistically. This may lead to problems. For instance, this does not take into account the redundancy of citations. Let's say a response has 4 aspects that need to be verified and 4 citations that all are duplicates of the same Web page. Although these citations verify only one aspect of the response, they will result in perfect citation recall and precision, according to the definitions of these metrics proposed in the paper. More fine-grained definitions would be more accurate. For instance, a response can only result in a citation recall of 1 if all aspects of it are covered by citations. Duplicate citations that partially support a response should not be counted towards precision.
* Some far-fetched and misleading conclusions: it seems quite counter-intuitive to me that close paraphrasing or copying of the source necessarily leads to lower fluency and perceived utility. It is completely unclear why paraphrasing leads to lower fluency (the response of Bing Chat in Figure 3 looks perfectly fluent to me). The relationship with perceived utility is indirect and rather depends on the relevance of the retrieved source text rather than whether it was copied or not. This problem is caused by the authors not providing any details or examples of what "fluent" and "cohesive" response is in the annotator guidelines.

**Reproducibility:**

2: Would be hard pressed to reproduce the results. The contribution depends on data that are simply not available outside the author's institution or consortium; not enough details are provided.

**Reviewer Confidence:**

4: Quite sure. I tried to check the important points carefully. It's unlikely, though conceivable, that I missed something that should affect my ratings.

**Typos Grammar Style And Presentation Improvements:**

according to the definition in section 2.4, the citation precision for the first response in Figure 2 should be 5/8, not 3/8
the authors should clearly associate numeric scores with different levels of the qualitative Likert scale for fluency and perceived utility, instead of having the readers guess

---

> ### Author Rebuttal · Authors · 2023-08-23
>
> Thanks for your thoughtful review. See below for some inline responses.
>
> ## Clearing up how we calculate citation precision / recall
>
> We think there might be a misunderstanding about how we define and compute the citation precision and recall metrics. Citation precision and recall are designed to behave exactly as you mention---in a nutshell, citation recall is 1.0 when every sentence is fully supported by their associated citations, and citation precision is 1.0 when every sentence contributes support for its associated sentence. Our metrics are calculated for a single response, but they rely on judgements on individual **sentences** in that response and each sentence's corresponding citations---the evaluation is sentence-level, but aggregated to give a response-level score.
>
> > Weaknesses in the definition of citation precision: although all citations for a single response are evaluated by a single worker, the citations are evaluated individually rather than holistically.
>
> We aren't quite sure what you mean by holistically, but note that we ask for 2 types of judgments in calculating citation precision:
> - For each _citation_, we ask if it (a) fully supports, (b) partially supports, or (c) does not support its associated sentence (L214 - L226)
> - For each _sentence_, we ask if it is (a) fully supported or (b) not fully supported by **the union of its associated citations** (L227 - L233)
>     - We believe that this is what you might be asking for when you say "holistically"
>
> > For instance, this does not take into account the redundancy of citations. Let's say a response has 4 aspects that need to be verified and 4 citations that all are duplicates of the same Web page. Although these citations verify only one aspect of the response, they will result in perfect citation recall and precision, according to the definitions of these metrics proposed in the paper.
>
> Our metric does take into account this redundancy. To make your example above concrete, let's say that we have a response with the following 4 sentences, and there are 4 citations that are all duplicates of the same webpage:
>
> - Sentence 1: Mice eat cheese [1].
> - Sentence 2: Mice eat bread [1].
> - Sentence 3: Mice drink coffee [1].
> - Sentence 4: Mice drink water [1].
>
> Let's say that citation [1] simply states "Mice eat cheese". Thus, it fully supports sentence 1, but provides no support for sentences 2, 3, 4.
>
> Let's first calculate citation recall. At a high level, citation recall asks: "what proportion of generated sentences are fully supported by their associated citations?". All 4 sentences here that are verification-worthy (L175). Now, for each of the 4 sentences, we evaluate "Evaluating whether a verification-worthy sentence is fully supported by (the union of) its associated citations". In this case, clearly only Sentence 1 is fully supported by its associated citation [1], while Sentences 2, 3, 4 are not. **Thus, since there is only 1 sentence that is fully supported by the union of its associated citations, the citation recall is 1/4 = 0.25, not 1.0**
>
> Now, let's calculate citation precision. At a high level, citation precision asks: "when you output a citation, what % of the time does that citation actually support the associated sentence?" To measure citation precision, we first ask annotators to rate, **for every outputted citation**, whether it fully, partially, or does not support its associated sentence. To be concrete, there are 4 outputted citations here: (a) [1] for sentence 1, (b) [1] for sentence 2, (c) [1] for sentence 3, (d) [1] for sentence 4 (since it's all just the same webpage). Of these 4 citations, only (a) ([1] for sentence 1) fully supports the associated sentence. The other 3 citations do not support their associated sentence. Thus, according to the definition of the metric in L234, $T_{fs} = 1$, $T_{ps} = 0$ (the partial support case does not apply here, since no citation partially supports its associated sentence), and $N = 4$ (since we outputted 4 citations). **Thus, precision is (1+0)/4 = 0.25, not 1.0**
>
> > More fine-grained definitions would be more accurate.
>
> Our evaluation is done at the sentence-level---there is currently no linguistic consensus on a definition for a sub-sentential "claim" or "statement", so we used sentences for reproducibility.
>
> > For instance, a response can only result in a citation recall of 1 if all aspects of it are covered by citations. Duplicate citations that partially support a response should not be counted towards precision.
>
> This is indeed how the metric is defined and designed.
>
> > according to the definition in section 2.4, the citation precision for the first response in Figure 2 should be 5/8, not 3/8
>
> **No, actually, according to the definition in section 2.4, the citation precision is indeed 3/8**. Here's a walk-through of the calculation: there are 8 outputted citations in total (3 for the first generated statement, 3 for the second generated statement, and 2 for the third generated statement), so $N = 8$. Now, we need to calculate $T_{fs}$, the proportion of citations that _fully support_ their associated statements:
>
> - Statement 1: citation [1] fully supports, citation [2] does not support, and citation [3] partially supports. So, we add 1 to $T_{fs}$ to get $T_{fs} = 1$
> - Statement 2: citation [1] fully supports, citation [2] does not support, and citation [4] does not support. So, we add 1 to $T_{fs}$ to get $T_{fs} = 1 + 1 = 2$
> - Statement 2: citation [4] fully supports, citation [5] partially supports. So, we add 1 to $T_{fs}$ to get $T_{fs} = 1 + 1 + 1 = 3$.
>
> So, in total, $T_{fs} = 3$. Now, let's calculate $T_{ps}$. As a reminder, we only count a citation toward $T_{ps}$ if (condition 1) it partially supports the associated statement, (condition 2) the associated statement is fully supported by the union of its associated citations, and (condition 3) the citation associated with this statement singlehandedly fully supports this statement. This might seem convoluted, but for an intuitive explanation, see footnote 4 on page 4 of the paper---the tl;dr is that we don't want to penalize systems for (correctly) aggregating information across multiple webpages. Ok, so let's calculate $T_{ps}$. As a reminder, all 3 conditions above need to hold to count a citation toward $T_{ps}$:
>
> - Statement 1: citation [3] partially supports (condition 1), but citation [1] fully supports the statement by itself (fails condition 2). Thus, citation [3] is actually redundant here---there's no point to having it, since [1] already covers it. So, we don't give the system credit for outputting this citation. $T_{ps} = 0$
> - Statement 2: None of the citations partially support, so $T_{ps} = 0$ remains 0.
> - Statement 3: citation [5] partially supports (condition 1), but citation [4] fully supports the statement by itself (fails condition 2). Thus, like in statement 1, citation [5] is actually redundant here---there's no point to having it, since [4] already covers it. So, we don't give the system credit for outputting this citation. $T_{ps} = 0$
>
> So, we have that $T_{ps} = 0$. **Thus, precision = $\frac{T_{fs} + T_{ps}}{N} = \frac{3 + 0}{8}$ = 3/8**
>
> ### Other comments:
>
> > Some far-fetched and misleading conclusions: it seems quite counter-intuitive to me that close paraphrasing or copying of the source necessarily leads to lower fluency and perceived utility.
>
> Note that just because a claim is counter-intuitive does not make it "far-fetched and misleading"---we also found these results to be somewhat surprising.
>
> > It is completely unclear why paraphrasing leads to lower fluency (the response of Bing Chat in Figure 3 looks perfectly fluent to me).
>
> Indeed, the correlation with fluency is weaker, and fluency is high overall among all systems. In the camera-ready version of the paper we're planning on focusing only on the relationship between precision and perceived utility and cutting the 3 other comparisons.
>
> > The relationship with perceived utility is indirect and rather depends on the relevance of the retrieved source text rather than whether it was copied or not.
>
> The (ir)relevance of the retrieved source text directly affects citation precision (increasing it) and perceived utility (decreasing it) when irrelevant text is copied. To be concrete, take Figure 3. The query asks about societal evolution, but one of the retrieved source texts is irrelevant because it talks about biological evolution (citation [1]). Nonetheless, the system copies directly from this citation. Since statement 1 is directly copied from the citation [1], citation [1] is almost guaranteed to fully support statement 1, which increases precision. However, since [1] is irrelevant to the query, this decreases perceived utility. This copying behavior is shown in a variety of responses (Section 4.4 / Table 2), and leads to the observed negative correlation between citation precision and perceived utility.
>
> > This problem is caused by the authors not providing any details or examples of what "fluent" and "cohesive" response is in the annotator guidelines.
>
> We're not sure that providing details or examples of what constitutes fluency / cohesive would have fixed this problem---collecting Likert scale judgments for generated text fluency and utility is a fairly standard practice.

---

### Meta-Review · Area_Chair_cZgQ · 2023-09-17

**Recommendation:** 3

**Metareview:**

This paper develops metrics to evaluate citations from generative search engines. They perform a human evaluation to determine the efficacy of four such search engines, and find that many statements of these search engines are not supported by citations. The released annotations will be useful for future research.

The primary concern following the discussion period is the lack of a method to quantify support of individual claims within a sentence, as mentioned by reviewer qPBo. There are additionally remarks from reviewer hMXQ about how the search engines may change; however, even if they change, the metrics can be used to evaluate their progress over time.

---

### Decision · Program_Chairs · 2023-10-07

**Decision:**

Accept-Findings

**Comment:**

This paper develops metrics to evaluate citations from generative search engines. They perform a human evaluation to determine the efficacy of four such search engines, and find that many statements of these search engines are not supported by citations. The released annotations will be useful for future research.

The primary concern following the discussion period is the lack of a method to quantify support of individual claims within a sentence, as mentioned by reviewer qPBo. There are additionally remarks from reviewer hMXQ about how the search engines may change; however, even if they change, the metrics can be used to evaluate their progress over time.